# Polyelectrolytes for Enzyme Immobilization and the Regulation of Their Properties

**DOI:** 10.3390/polym14194204

**Published:** 2022-10-07

**Authors:** Vladimir I. Muronetz, Denis V. Pozdyshev, Pavel I. Semenyuk

**Affiliations:** 1Belozersky Research Institute of Physico-Chemical Biology, Lomonosov Moscow State University, Leninskie Gory 1, Bld 40, 119992 Moscow, Russia; 2Butlerov Chemical Institute, Kazan Federal University, Kremlevskaya 18, 420008 Kazan, Russia

**Keywords:** smart polymers, enzyme immobilization, polyelectrolytes, polyanions, polycations, artificial chaperones, inclusion bodies, protein aggregation

## Abstract

In this review, we considered aspects related to the application of polyelectrolytes, primarily synthetic polyanions and polycations, to immobilize enzymes and regulate their properties. We mainly focused on the description of works in which polyelectrolytes were used to create complex and unusual systems (self-regulated enzyme–polyelectrolyte complexes, artificial chaperones, polyelectrolyte brushes, layer-by-layer immobilization and others). These works represent the field of “smart polymers”, whilst the trivial use of charged polymers as carriers for adsorption or covalent immobilization of proteins is beyond the scope of this short review. In addition, we have included a section on the molecular modeling of interactions between proteins and polyelectrolytes, as modeling the binding of proteins with a strictly defined, and already known, spatial structure, to disordered polymeric molecules has its own unique characteristics.

## 1. Introduction

Polyelectrolytes, both natural and synthetic, are widely used for enzyme immobilization. As a rule, charged matrices are needed for the physical adsorption of enzymes with opposite charges. With the help of such simple immobilization, it is possible to obtain preparations of insoluble enzymes, as well as solve other problems, by creating a favorable microenvironment for the enzyme, increasing the accumulation of charged substrates near the adsorbed protein and so on. Thousands of articles and dozens of reviews are devoted to these aspects of enzyme immobilization on polyelectrolytes. In this article, we would like to focus on those works in which the unique properties of polyelectrolyte systems are used to immobilize enzymes and regulate their properties. These properties include the possibility of obtaining particles consisting of oppositely charged polyelectrolytes. By adjusting the composition of such particles, the size and charge of polyelectrolytes, as well as the ionic strength and pH of the medium, it is possible to obtain water-soluble or insoluble particles of different sizes and with a variety of structures. In this way, it is possible to create unique systems that are similar to natural systems in terms of their complexity and ability to self-regulate, such as chaperones.

The choice of the review topic is also related to the fact that I would like to highlight in detail the work of the outstanding specialist in the field of polyelectrolytes, Prof. Vladimir Izumrudov. Our friend and colleague Vladimir Izumrudov passed away in February 2022. For half a century, in cooperation with enzymologists, biochemists and molecular biologists, Vladimir Izumrudov obtained extremely important and original results related to the use of polyelectrolytes, including those synthesized by himself, to solve various problems in biotechnology and physicochemical biology.

In this review, we will consider the use of polyelectrolyte systems for enzyme immobilization in detail and for creating systems with controlled solubility, including self-regulating solubility that changes during the reaction. A separate section will be devoted to creating artificial chaperones based on polyelectrolytes, from the simplest, similar to small heat shock proteins, to more complex ones that can not only bind to denatured proteins, but also recognize unfolded polypeptide chains and mimic chaperonin functional cycles. In addition, the specifics of molecular modeling of interactions of polyelectrolytes with proteins will be considered. The synthesis and stability of the polyelectrolytes and the basic properties of protein–polyelectrolyte complexes are both beyond the scope of this review, and they are described in many excellent comprehensive reviews [1,2,3,4,5,6,7].

## 2. Polyelectrolyte Complexes with Immobilized Enzymes for Obtaining Self-Regulating Biocatalysts

Positively or negatively charged polyelectrolytes are widely used for the adsorption or covalent immobilization of enzymes, which makes it possible to influence both the properties of the enzymes themselves and their catalytic processes. However, the preparation of complexes of oppositely charged polyelectrolytes has opened up completely new possibilities for creating polyelectrolyte-based biocatalysts [8,9].

Polyelectrolyte complexes have been intensively studied since the late 1970s [10,11,12]. It was found that water-soluble particles of negatively and positively charged polyelectrolytes can be obtained at a certain ratio. At the same time, insoluble particles can be obtained by either changing the ratio of the two polyelectrolytes or the composition of the medium (pH or ionic strength). The transition to an insoluble state can occur with a relatively small change in pH or ionic strength. In particular, non-stoichiometric water-soluble complexes of poly(4-vinyl-N- ethylpyridinium bromide) and poly(methacrylic acid) in a ratio of 1:3, were obtained, which formed particles larger than one million daltons at neutral pH values (pH 7.5, 0.1 M NaCl). In such particles, the core is formed due to ionic bonds between oppositely charged polyelectrolytes, and the hydrophilic shell is formed due to the ionized groups of the polyelectrolyte that are present in excess. When the medium is acidified, even by 0.2 units (or the ionic strength is changed), the polyelectrolyte complexes precipitate out due to a decrease in the total density of negative charges [10,11,12].

Based on these observations, the idea arose to include enzymes in these polyelectrolyte complexes, since it is very important to be able to regulate the solubility in water in the preparation of biocatalysts. In a series of works, a number of enzymes (penicillin amidase, alcohol dehydrogenase) were immobilized covalently on positively charged poly(4-vinyl-N-ethylpyridinium bromide), and the water-soluble polyelectrolyte complexes described above were obtained using poly(methacrylic acid) as the second polymer [13,14,15,16,17]. In the first work, penicillin amidase was immobilized on poly(4-vinyl-N-ethylpyridinium bromide), and polyelectrolyte complexes were obtained [13]. The kinetic parameters of the substrate hydrolysis changed insignificantly, although there was an expansion of the pH dependence of the activity and a change in some of the other properties of the enzyme due to the characteristics of the microenvironment. The most interesting observation was that acidification or a change in the ionic strength of the medium led to the transition of the polyelectrolyte particles into an insoluble state and, consequently, to the termination of the enzymatic reaction in the incubation medium. In this case, after the alkalization of the medium to the initial pH values, the polyelectrolyte particles become soluble again, and the enzyme activity is completely restored.

Of course, this effect was interesting in itself, since it made it possible to stop the enzymatic reaction and initiate it again if necessary. Based on such systems, it was possible to obtain multiple-acting biocatalysts by separating them from the incubation medium. However, even more interesting is the concept of the creation of self-regulating biocatalysts proposed by the authors [13], shown in Figure 1.

In the first stage, the enzyme immobilized on polyelectrolyte particles catalyzes a reaction resulting in a change in pH and/or ionic strength of the medium due to product accumulation. In the next stage, such changes lead to the transition of the complexes to an insoluble state and a slowdown or termination of the enzymatic reaction. The subsequent utilization of the reaction products (for example, by coupled enzymes) or the addition of new portions of the substrate alkalizing the medium, causes the transition of the complexes to a soluble state and the resumption of the enzymatic reaction. Unfortunately, the proposed original and very elegant model has not been developed, although many systems, such as those based on temperature-controlled polymers, exploit the same idea. In our opinion, it was these works that laid the foundation for a whole trend related to the use of “smart” polymers in biocatalysis [18,19,20].

## 3. Immobilization of Enzyme via Polymers

Polyelectrolytes are widely used for non-covalent enzyme immobilization. Among the numerous approaches suggested, layer-by-layer immobilization seems to be one of the most common. It consists of sequential absorption of polyanionic and polycationic layers using electrostatic forces [21]. The thickness of the film produced can vary from a few to a dozen layers or more; the target enzyme can represent the last layer, or alternatively, the enzyme (or several different enzymes) can be absorbed onto the inner layers [22]. This approach enables the construction of complex systems with multiple enzymatic activities, such as bioreactors [23], biosensors [24], biofuel cells [25] or other surface functionalization [26]. The use of charged polypeptides [27,28] or hydrolyzed polymers [29] as a polycationic or polyanionic component for layer-by-layer immobilization is promising in terms of biodegradability and biosustainability.

Polyelectrolyte brushes are also widely used for protein immobilization [30,31]. In this approach, polymer chains are bound to the surface by one of the ends and non-covalently sorb oppositely charged proteins with the outer part. This provides a high loading of an active enzyme [32]. In addition, a broad spectrum of technologies, such as photolithography, enables powerful functionalization of the target surface with complex patterns and other tuning capabilities [33,34].

For protein immobilization on small soluble particles or nanoparticles, a simple complexation with linear or branched polyelectrolytes or dendrimers may be the best choice [9,35,36]. The behavior of such complexes strongly depends on conditions and the molar ratio of protein and polymer, varying from the formation of large, precipitated aggregates to stable, soluble complexes [3,8], thus providing an excellent opportunity for manipulation and reversible precipitation–solubilization of the complex. The use of copolymers enables stabilization of the particles in a broad range of conditions through micelle formation; the polyelectrolyte moiety of the copolymer is responsible for protein binding, whereas the other part, for example, a non-ionic polymer such as poly(ethylene oxide), provides solubility of the complex/micelle [37,38,39]. In the mentioned cases, the protein–polyelectrolyte complex represents an interpolyelectrolyte complex in which protein serves as a polycation or polyanion. In addition, non-protein interpolyelectrolyte complexes can capture proteins during formation, as was used, for example, to immobilize an antigen on a polymer-based carrier, and thus enhance vaccine efficiency in [40].

Polymeric microgels and nanogels are also a useful way for protein immobilization [41]. The microgel can be made from one or two (or sometimes a few) different polymers (polyelectrolytes), usually cross-linked to each other. The main advantage of this approach is an ability to swell and shrink under different conditions; being cross-linked covalently, the polymer chains stay tied instead of undergoing a complete dissociation of the complex. It enables sorption followed by an efficient release of the bound enzyme under specific conditions instead of a permanent immobilization. The swelling/shrinking (and therefore release of the absorbed enzyme) can be controlled by temperature, salt, light or other stimuli [42,43,44,45].

Finally, polyelectrolytes can be used as a part of more complex nanoparticles. For example, polycations were used as a “glue” to immobilize an anionic epitope on the tobacco mosaic virus particles used as a core for antigen-carrying nanoparticles [46].

In summary, polyelectrolytes are a powerful tool for the immobilization of enzymes or other proteins on surfaces or nanoparticles. A strong interaction capability, both between polycations and polyanions and between protein and polyelectrolytes, provides a good immobilization outcome without covalent binding, which would require additional reagents and sometimes complicated washing techniques. On the other hand, non-covalent binding enables various opportunities for regulation: the immobilized enzyme can be released under the desired conditions.

## 4. Effect of Immobilization on Enzyme Activity

Immobilization on polyelectrolytes involves strong interactions between the enzyme and polymer and consequently, might influence protein structure and activity. Thus, sulfated hydrophobic polymers, whose high affinity to proteins is determined by sulfated groups and significantly corroborated with hydrophobic interactions, are known to induce significant changes in protein structure and protein inactivation [47,48,49]. This is a strong argument in favor of using weak polyelectrolytes such as poly(acrylic acid) or other polycarboxylates and polyamines for immobilization. However, such destructive action might be reversible, suggesting a useful tool for enzymatic activity manipulation; controlled binding and release of the enzyme enables control of its activity [50,51].

Sometimes immobilization might enhance the activity of the enzyme. Other examples represent a successful combination of an enzyme, its substrate, and a polymer. Thus, chymotrypsin was hyperactivated after complexation with a polymer of a charge complementary to a charge of the substrate [52]. A similar effect was demonstrated for another protease, subtilisin, probably via enhanced binding of calcium cations (important for catalysis) after enzyme complexation with polyanions [53].

In summary, immobilization might help to stabilize the enzyme or even destabilize it, in the case of suboptimal polymer selection. Sometimes it may represent an opportunity for additional activation of the enzyme.

## 5. Application of Polyelectrolytes and Polyelectrolyte Complexes as Artificial Chaperones

The use of enzymes as biocatalysts is complicated by their low stability. There are several ways to solve this problem: the use of stable enzymes (for example, from heat-resistant microorganisms); the stabilization of enzymes, including the use of cross-linking reagents or immobilization; or the production of stable enzymes by genetic engineering. In addition, denatured forms of proteins can be reactivated. Reactivation is also often required for recombinant proteins isolated in the inactive state as inclusion bodies. A variety of chaperones can be used to reactivate proteins, from the simplest small heat shock proteins to complex ATP-dependent chaperonins. Chaperones are used both for the reactivation of enzymes in vitro and for their expression in producing microorganisms simultaneously with the expression of target proteins [54].

However, chaperones, particularly complex chaperonins, are not easy to isolate. The ATP-dependent nature of their functioning also complicates the reactivation process. For these reasons, many attempts have been made to use polyelectrolytes or polyelectrolyte complexes as artificial chaperones. The easiest approach is to mimic the action of the simplest ATP-independent chaperones, such as small heat shock proteins. Such chaperones effectively prevent protein aggregation and can also destroy already formed aggregates. In fact, they are natural amphiphilic polyelectrolytes bearing additional hydrophobic regions. Enzyme aggregation usually occurs at high protein concentrations and can be prevented simply by binding an excess amount of polypeptide chains. This effect can be easily achieved by adding a negatively or positively charged polyelectrolyte to the system where the enzyme is reactivated. Such experiments have been successfully carried out in a number of laboratories, and it was possible to enhance the effect by introducing hydrophobic units into the polyelectrolyte [55,56].

The binding of part of the unfolded polypeptide chains with polyelectrolytes prevented their interaction with each other and prevented aggregation due to a decrease in the protein concentration in the solution. This increased the yield of reactivated active enzyme. However, removing a portion of the protein from the reactivation process is disadvantageous since the same effect can be achieved simply by lowering the concentration of the reactivated protein. For this reason, the next stage of the use of polyelectrolytes as artificial chaperones included the release of proteins associated with them. First, part of the unfolded polypeptide chains was bound to a positively or negatively charged polyelectrolyte, and then the unbound polypeptide chains were renatured. After the separation of the polyelectrolyte complex with the unfolded polypeptide chains from the renatured protein, the dissociation of the complex and the release of polypeptide chains into solution with subsequent reactivation was initiated. Such a transition could be achieved by changing the pH value or increasing the ionic strength of the medium [39,57,58,59]. However, the most promising option is the addition of an oppositely charged polyelectrolyte to the complex [60]. This makes it possible to displace the polypeptide chains from the complex with the polyelectrolyte, simultaneously release the enzyme into the solution and remove two oppositely charged polyelectrolytes by precipitation. This approach worked both to isolate the enzyme from the complex with the polyanion by adding the polycation [57,61] (Figure 2), and to displace the enzyme from the complex with the polycation by adding the polyanion [62,63].

It is important that in the examples listed and in other works [64], the released enzyme can renature spontaneously in the solution, which makes the restoration of its activity possible. It should be noted that it is possible to create a pulsating system in which the transition of polypeptide chains into the solution will occur in small portions to prevent their aggregation.

However, the experiments described above imitate only one side of the action of the chaperone: binding to the substrate. This prevents the aggregation of polypeptide chains, performing the so-called holdase chaperone-like activity. At the same time, the binding remains nonspecific, as it is usually determined only by the polyelectrolyte and the polypeptide chain charge. However, sometimes this is sufficient to achieve a certain binding specificity and even isolate one of the proteins from a multicomponent [65,66,67]. From this point of view, it seems promising to use more complex polymers, for example, thermosensitive polymers. The binding of such molecules to a protein depends on temperature, which makes it possible to recognize and bind only the unfolded state of the enzyme without affecting the native active form of the enzyme [68,69].

However, natural molecular chaperones, especially complex chaperonins, often only recognize specific structural motifs of polypeptide chains. Moreover, chaperonins, especially eukaryotic ones such as TRiC, are only able to recognize unfolded polypeptide chains of certain substrate proteins, which indicates a rather high substrate specificity. Therefore, in order to bring the characteristics of the polyelectrolyte-based artificial chaperones closer to complex chaperonins, it is necessary to introduce some additional elements into the polyelectrolytes that only recognize certain structures. Naturally, antibodies are most suitable for this role since they have a high affinity for certain antigenic determinants. Such antigenic determinants can be elements of the protein structure that are characteristic of its denatured forms alone. It is not a very difficult task to obtain antibodies that selectively bind to non-native forms of proteins, but do not interact with native proteins [70]. Monoclonal antibodies can be obtained by selecting clones for unfolded polypeptide chains [71]. Such monoclonal antibodies are often used to identify proteins after electrophoresis under denaturing conditions. In addition, antibodies to non-native forms of a protein can be isolated from polyclonal antisera using affinity chromatography on carriers with immobilized native or unfolded antigens [72].

The first attempts to obtain artificial chaperones, based on the antibodies immobilized on polyelectrolytes, were carried out using monoclonal antibodies that interact with different forms of glyceraldehyde-3-phosphate dehydrogenase, but do not recognize its native form [71]. The antibodies to non-native forms of the enzyme were covalently immobilized on poly(methacrylic acid) that forms a polyelectrolyte complex with the polycation, poly-(N-ethyl-4-vinylpyridinium bromide) [73,74]. Such complexes could be easily precipitated from the solution by changing the pH or ionic strength of the medium. The addition of the polyelectrolyte complex to the denatured enzyme during its reactivation led to an increase in its specific activity. This effect was due to the binding of denatured forms of the protein to the monoclonal antibodies conjugated with the complex. On the one hand, the removal of part of the protein from the reactivation mixture decreased protein aggregation. On the other hand, the removal of misfolded forms of the protein by precipitation of the polyelectrolyte complex increased the specific activity of the reactivated enzyme in the solution (Figure 3).

The use of antibodies conjugated with polyelectrolyte complexes increased the specificity of such artificial chaperones. However, this did not produce a system that imitated the action of natural chaperones, in which the balance between unfolded and folded forms of proteins is maintained. An attempt to create such a system was made using glyceraldehyde-3-phosphate dehydrogenase and monoclonal antibodies to non-native forms of this enzyme [75]. The antibodies and denatured glyceraldehyde-3-phosphatedehydrogenase were covalently linked to different chains of poly(methacrylic acid). The polyanion chains only interacted by the binding of the antibodies to their antigen, and it was possible to transfer the polyelectrolyte particles from a soluble state to an insoluble one and also to change the structure of the complexes by changing the pH value. The addition of denatured forms of the enzyme to such a system resulted in an equilibrium between bound and soluble polypeptide chains of the enzyme. After the initiation of reactivation of the denatured enzyme by dilution, the concentration of the denatured protein forms in the solution decreased, which led to the release of new portions of the denatured protein from the complex with immobilized antibodies into the solution. Thus, the reactivation of the enzyme in solution occurred at a low protein concentration of the denatured enzyme, which prevented its aggregation.

Thus, using polyelectrolyte complexes with the immobilized antibodies to non-native forms of the enzyme, both the high specificity of chaperone action and their ability to regulate the balance between native and denatured forms of proteins can be simulated. A pulsating change in the pH or ionic strength of the medium, leading to a change in the binding of polyelectrolytes to different forms of proteins or a change in the solubility of complexes, can replace the cyclic action of chaperones achieved by ATP-dependent processes.

Separately, it is worth noting the possibility of not only inhibiting the aggregation of enzymes due to their immobilization on polyelectrolytes, but also the disassembly of already formed aggregates. In particular, the dissolution of inclusion bodies and the subsequent release of the folded enzyme in a free form is an actual problem for bioengineering and biotechnology. Using polyelectrolytes, it is possible to destroy both inclusion bodies [76] and other types of aggregates, including strong amyloid fibrils [77,78,79]. This approach can be convenient for the immobilization of proteins on polymers, for obtaining an active enzyme and for recycling the “spoiled” fraction back to the reaction mixture, in the case of the subsequent release of the protein from the polymer complex.

## 6. Modeling of the Complexation

Molecular modeling is a powerful tool for studying the binding between proteins and polymers. The first question that can be answered with the use of simulations is the binding mechanism. Since proteins and most of the polymers used for immobilization share an amphipathic nature, the impact of particular types of interactions is not trivial. In a set of excellent papers that have become classics in protein–polymer science, M. Ballauff’s group elucidated the mechanism of the binding and demonstrated the impact of entropy gain arising from counter-ion release in the protein–polyelectrolyte interaction [80,81,82,83]. Although clear experimental evidence, generally from isothermal titration calorimetry, was shown, molecular modeling, including coarse-grained simulations, made a significant contribution to this study.

Second, binding affinity or specificity for sites on the protein surface might be predicted in silico, enabling a screening study, which would be complicated to perform in vitro. This idea can be exemplified by numerous studies of sulfated polysaccharides (such as glycosaminoglycans) binding with proteins [84,85,86,87]. In addition to binding site evaluation, different polymers (including similar glycosaminoglycans with different sulfation patterns) might be compared for the binding efficiency under particular conditions. Protein-based nanoparticles or other protein immobilization systems might be designed using modeling approaches [67,88].

Molecular modeling can be useful for studying the effect of polyelectrolyte binding on protein structure and behavior [89]. Modeling loops and tails formed by the polymer chain bound to protein enabled the stability of the complex and chaperone-like activity of the polyelectrolytes to be estimated [90,91]. It is of special importance, not only for the immobilization of enzymes but also for the investigation of potential anti-amyloid action of polyelectrolytes [92,93,94], especially in light of the possible participation of natural polysaccharides in amyloidosis.

It is noteworthy that computational approaches enable the testing of the effect of protein point mutations on its binding to a polyelectrolyte [95,96], providing a good opportunity for in silico estimation of the prospect of the modified enzyme immobilization. It might also be useful for studying the biological effects of post-translational modification, such as glycosylation [97,98] since cells contain many types of non-protein polyelectrolytes [99].

In summary, molecular modeling, especially atomistic simulations, enables the prediction of: (1) the binding affinity; (2) the effect of binding on enzyme structure and activity; (3) the overall composition of the complex or nanoparticle. The estimation of the binding affinity enables the screening of various polymers and different protein forms or point mutations to obtain the best complexation. As a result, molecular modeling of such immobilized systems should be a key method of future protein engineering. Indeed, in addition to conventional approaches, such as introducing point mutations to enhance protein stability or increase its activity, immobilization on polyelectrolytes might help alter or boost the enzyme activity. This approach requires fine-tuning an enzyme, a polyelectrolyte and the complex, which cannot be done without molecular modeling.

## 7. Conclusions and Perspectives

Polyelectrolytes can be used to immobilize enzymes and create systems to regulate enzyme properties (catalytic parameters and stability), reactivate denatured proteins, and prevent their aggregation. The diverse application of polyelectrolytes for the use of enzymes in biotechnology justifies the emergence of a whole field called “smart polymers”. However, not all the potential of synthetic and natural polyelectrolytes has been used to produce enzymes with unusual properties. The examples given in the review show that complexes of polyelectrolytes with proteins allow many problems in biotechnology to be solved, and with the further development of this field, it will find increasing application in other areas, primarily in biological and medical sciences.

A strong binding, followed by partial unfolding and inactivation of the bound enzyme, might be disadvantageous. The low specificity and the potential reactivity of polyelectrolytes towards other proteins that might result in side effects in complicated systems oriented to biotechnology and medicine can be considered a disadvantage. However, these disadvantages can be solved with a careful selection of polyelectrolytes optimal for a particular system or with the construction of complex systems, such as those containing stimuli-responsive polymers or antibodies, for a high level of specificity. Some examples of the immobilization systems are listed in Table 1.

Polyelectrolyte-based immobilization of enzymes enables a powerful manipulation of enzyme activity and recognition of particular states/forms of proteins in line with the idea of smart polymers. This technology (or set of technologies) has great potential for practical use in biotechnology, bioengineering, and medicine that demands protein engineering to obtain the desired activity and features. In addition to direct immobilization, polyelectrolytes might boost enzyme functionality and help to regulate its behavior, representing a direction for future research and development. Furthermore, polyelectrolyte-based protein immobilization and complexation using biodegradable or natural polymers are suitable for biosustainable applications.

## Figures and Tables

**Figure 1 polymers-14-04204-f001:**
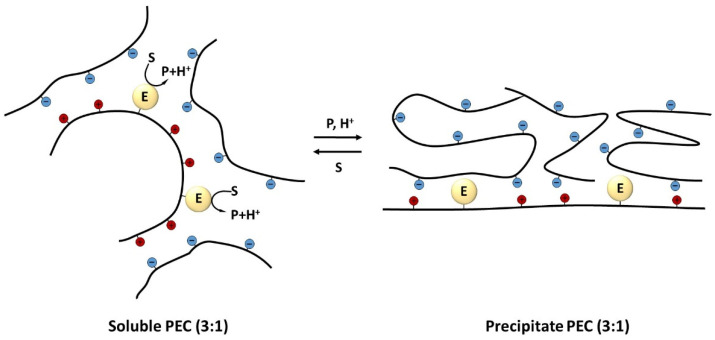
Self-regulated biocatalysts based on enzymes immobilized on polyelectrolytes (PEC—polyelectrolyte complex) (modified from [13]). An enzyme immobilized on polycation (E) catalyzes a reaction (S—substrate, P—product) that increases the H^+^ concentration in the system. The polyanion (threefold excess to polycation) acidification leads to the PEC negative charge decrease, polyelectrolyte chains folding and PEC precipitation.

**Figure 2 polymers-14-04204-f002:**
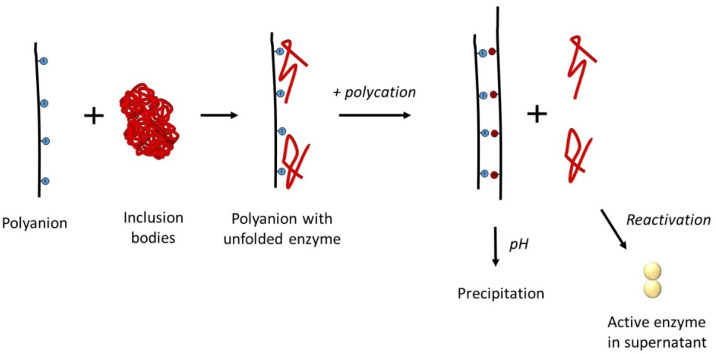
Reactivation of enzymes during their release into solution by the interaction of oppositely charged polyelectrolytes.

**Figure 3 polymers-14-04204-f003:**
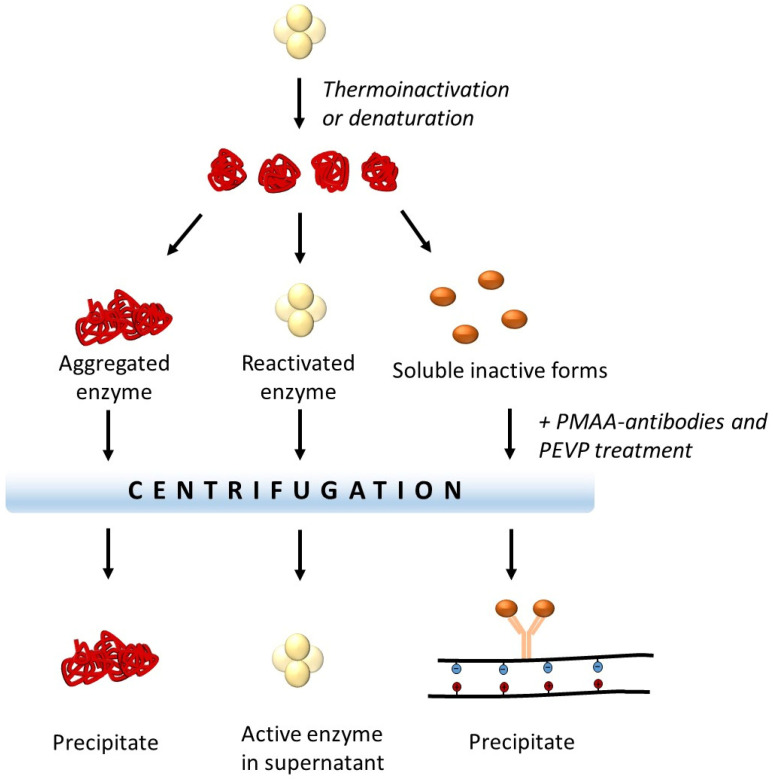
The use of polyelectrolytes with immobilized antibodies against non-native forms of the protein to increase the specific activity of enzymes (modified from the article [74]) PEVP—poly(4-vinyl-N-ethylpyridinium bromide), PMAA—poly(methacrylic acid).

**Table 1 polymers-14-04204-t001:** Examples of different immobilization approaches.

Method of Immobilization	Polyelectrolyte	References
Covalent binding of enzyme or antibody	poly(4-vinyl-N-ethylpyridinium bromide), poly(methacrylic acid), poly(acrylic acid)	[13,14,15,16,17]
Layer-by-layer	poly(styrenesulfonate), poly(ethyleneimine), poly(dimethyldiallylammonium)	[23,24,25,26]
Electrostatic binding to polyanion or polycation	poly(acrylic acid), poly(methacrylic acid)	[32,52,53,58,60]
	poly[2-(methacryloyloxy)ethyl]trimethylammonium	[30]
	poly(diethylamino)methyl methacrylate	[32]
	poly(allylamine)	[52]
	poly(l-γ-glutamic acid)	[53]
	poly(2-aminoethylmethacrylate hydrochloride)	[59]
	Carboxymethylcellulose, poly(L-aspartate), poly(vinylsulfonate), heparin, dextran sulfate, poly(styrene sulfonate)	[60,63]
	poly(N,N-diethylaminoethyl methacrylate)-graft-poly(ethylene glycol)	[62]
	poly(allylurea-co-allylamine) and succinylated and acetylated derivatives	[66]
Incorporation into microgel	poly(N-isopropylacrylamide)	[42,43]
	poly(N-isopropylacrylamide-co-N-(3-dimethylaminopropyl)methacrylamide)	[45]
	poly(2-((2-(methacryloyloxy)ethyl)dimethylammonio)acetyl)(phenylsulfonyl)amide	[44]

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
