# Peer review of "Polyelectrolytes for Enzyme Immobilization and the Regulation of Their Properties"

_polymers, 2022, doi:10.3390/polym14194204_

Round 1
Reviewer 1 Report
This review aims to provide an overview of the roles of polyelectrolytes in enzyme immobilization and properties regulation. After going through the manuscript, I have some comments as follows:
11. Allow me to offer my condolences on your loss. Maybe the editor will consider the paragraph related to Prof. Vladimir Izumrudov. I am not sure it fits with the format of the journal.
22. The figure was presented as low resolution. So, their solution needs to be enhanced.
33. The English writing of the paper needs to be improved.
44. Different types of polyelectrolytes, their synthesis, stability, and performance need to be presented in this work.
55. The authors need to provide the reader with the summarizing table on all previous work on polyelectrolytes for enzyme immobilization.
66. The advantages and disadvantages of polyelectrolytes for enzyme immobilization need to be mentioned clearly.
77. The authors need to give a perspective on this direction of research.
Author Response
‘1. Allow me to offer my condolences on your loss. Maybe the editor will consider the paragraph related to Prof. Vladimir Izumrudov. I am not sure it fits with the format of the journal.’
Thank you for your condolences. We would have preferred that the phrase about Prof. Vladimir Izumrudov remain in the text. But will accept any editor's decision clearly.
‘2. The figure was presented as low resolution. So, their solution needs to be enhanced.’
The resolution of the figures has been corrected.
‘3. The English writing of the paper needs to be improved.’
The text has been re-reviewed.
‘4. Different types of polyelectrolytes, their synthesis, stability, and performance need to be presented in this work.’
We added a short comment to the text. According to the scope of the present review and our area of expertise, we briefly mentioned these topics and referenced excellent reviews by other authors who comprehensively describe the synthesis and properties of the polyelectrolytes.
‘5. The authors need to provide the reader with the summarizing table on all previous work on polyelectrolytes for enzyme immobilization.’
We added Table 1 with examples from the papers cited in the present review. The vast amount of papers on polyelectrolytes for enzyme immobilization, including conventional sorbents for protein immobilization, makes it difficult to summarize and cite them all.
‘6. The advantages and disadvantages of polyelectrolytes for enzyme immobilization need to be mentioned clearly.’
We added a paragraph about advantages and disadvantages to Conclusions section.
‘7. The authors need to give a perspective on this direction of research.’
We added a paragraph about perspectives to Conclusions section, which was renamed to “Conclusions and perspectives”.
Reviewer 2 Report
Dear Authors, It is a well-presented paper and easy to read. I have a few minor points to make. The text in the figures is too small, and so are the arrows that are too small, and barely visible. On page 8, there is a duplicate of 2 words in line 316. In the bibliographical references, I appreciated that the majority of the cited references have their doi. Perhaps a little too much self-citation, but there is nothing to complain about because these are very useful for this paper, and the bibliography is perfect. I appreciated the synthesis of the modeling which is well presented. I have a favorable opinion of this paper.
Author Response
‘Dear Authors, It is a well-presented paper and easy to read. I have a few minor points to make. The text in the figures is too small, and so are the arrows that are too small, and barely visible. On page 8, there is a duplicate of 2 words in line 316. In the bibliographical references, I appreciated that the majority of the cited references have their doi. Perhaps a little too much self-citation, but there is nothing to complain about because these are very useful for this paper, and the bibliography is perfect. I appreciated the synthesis of the modeling which is well presented. I have a favorable opinion of this paper.’
In accordance with the reviewer's comments, the figures were revised, and an error on line 316 was corrected.
Round 2
Reviewer 1 Report
The authors indicated my review's comments properly. So, I recommend this work for publication in the Polymer journal.